# HYBRID 3D-4D GAUSSIAN SPLATTING FOR FAST DYNAMIC SCENE REPRESENTATION

## ABSTRACT

Recent advancements in dynamic 3D scene reconstruction have shown promising results, enabling high-fidelity 3D novel view synthesis with improved temporal consistency. Among these, 4D Gaussian Splatting (4DGS) has emerged as an appealing approach due to its ability to model high-fidelity spatial and temporal variations. However, existing methods suffer from substantial computational and memory overhead due to the redundant allocation of 4D Gaussians to static regions, which can also degrade image quality. In this work, we introduce hybrid 3D–4D Gaussian Splatting (3D-4DGS), a novel framework that adaptively represents static regions with 3D Gaussians while reserving 4D Gaussians for dynamic elements. Our method begins with a fully 4D Gaussian representation and iteratively converts temporally invariant Gaussians into 3D, significantly reducing the number of parameters and improving computational efficiency. Meanwhile, dynamic Gaussians retain their full 4D representation, capturing complex motions with high fidelity. Our approach achieves significantly faster training times compared to baseline 4D Gaussian Splatting methods while maintaining or improving the visual quality.

## 1 INTRODUCTION

Accurately representing and rendering complex dynamic 3D scenes is fundamental to a wide range of applications, including immersive media for virtual and augmented reality. For example, in commercial and industrial domains such as sports broadcasting, film production, and live performances, the demand for high-quality dynamic scene reconstruction continues to grow, driven by the need for enhanced viewer engagement. While significant progress has been made, achieving high-fidelity, computationally efficient, and temporally coherent modeling of dynamic scenes remains a challenging problem.

Recent advances in neural rendering, particularly Neural Radiance Fields (NeRF) (Mildenhall et al., 2021; Barron et al., 2022; 2023; Fridovich-Keil et al., 2022; Sun et al., 2022a; Müller et al., 2022), have emerged as a powerful representation for novel view synthesis and 3D scene reconstruction, leveraging neural networks, grid-based data structures, and volumetric rendering (Brebin et al., 1998). Extensions of NeRF to dynamic 3D scene modeling (Song et al., 2023; Lombardi et al., 2019; Pumarola et al., 2021; Park et al., 2021a;b; Fridovich-Keil et al., 2023; Cao & Johnson, 2023; Wang et al., 2023; Mihajlovic et al., 2023; 2024; Kim et al., 2024) have shown promising results, enabling the reconstruction of time-varying environments with improved fidelity. However, real-time and high-fidelity rendering of complex dynamic scenes continues to be an open problem due to the computational cost of volume rendering and the complexity of spatio-temporal modeling.

More recently, 3D Gaussian Splatting (3DGS) (Kerbl et al., 2023) has become a promising alternative to NeRF-based approaches for 3D scene reconstruction and novel view synthesis , offering improved quality and real-time rendering capabilities. Unlike NeRF, which relies on implicit representation and computationally expensive volumetric rendering, 3DGS represents scenes as a collection of Gaussian primitives and leverages a fast rasterization. Several extensions have been proposed to adapt 3DGS for dynamic 3D scene reconstruction, incorporating motion modeling and temporal consistency to handle time-varying environments.

Two primary paradigms have been developed for applying 3DGS to dynamic 3D capture. The first approach *extends 3D Gaussians to dynamic 3D scenes* by tracking Gaussians over time (Li et al.,

2024; Wu et al., 2024; Lee et al., 2025a; Huang et al., 2024; Zhu et al., 2024; Kratimenos et al., 2024), using techniques such as multi-layer perceptrons (Li et al., 2024), temporal residuals (Wu et al., 2024), or interpolation functions (Lee et al., 2025a). These methods leverage temporal redundancy across frames to improve the representation efficiency and accelerate training, but they often struggle with fast-moving objects. The second paradigm, *directly optimizing 4D Gaussians*, represents the entire spatio-temporal volume as a set of splatted 4D Gaussians (Yang et al., 2023a; Duan et al., 2024; Lu et al., 2024a). While this approach enables high-quality reconstructions, it incurs significant memory and computational overhead. Furthermore, allocating 4D Gaussians to inherently static regions is inefficient, as these areas do not benefit from time-varying parameters (Cho et al., 2024).

In this work, we propose a *hybrid 3D-4D Gaussian Splatting (3D-4DGS)* framework that addresses the inefficiencies of conventional 4DGS pipelines. A key limitation of 4DGS (Yang et al., 2023a) is their treatment of static regions, which often requires multiple 4D Gaussians across different timesteps. While an optimal solution would involve assigning large scales along the temporal axis to represent static regions more effectively, this rarely occurs in practice. We propose a hybrid approach that models static regions with 3D Gaussians while reserving 4D Gaussians for dynamic elements. The proposed approach significantly reduces the number of Gaussians, leading to lower memory consumption and faster training speed.

Our approach begins by modeling all Gaussians as 4D and then adaptively identifying those with minimal temporal variation across the sequence. These Gaussians are classified as static and converted into a purely 3D representation by discarding the time dimension, effectively freezing their position, rotation, and color parameters. Meanwhile, fully dynamic Gaussians retain their 4D nature to capture complex motion. Importantly, this classification is not a one-time process but is performed iteratively at each densification stage, progressively refining the regions that truly require 4D modeling. The final rendering pipeline seamlessly integrates both 3D and 4D Gaussians, projecting them into screen space for alpha compositing. This design ensures that temporal modeling is applied where necessary, capturing motion effectively while eliminating redundant overhead in static regions.

We demonstrate the effectiveness of the proposed *3D-4DGS* on two standard challenging datasets: *Neural 3D Video (N3V)* (Li et al., 2022), which primarily comprises 10-second multi-view videos (plus one 40-second long sequence), and *Technicolor* (Sabater et al., 2017), featuring 16-camera light field captures of short but complex scenes. Our method consistently achieves competitive or superior PSNR and SSIM scores while significantly reducing training times. Additionally, we conduct ablation studies to reveal how key design choices—such as the scale threshold and opacity reset strategies—impact final quality and efficiency. We summarize our main contributions as follows:

- **Hybrid 3D–4D representation.** We introduce a novel approach, *3D-4DGS*, that dynamically classifies Gaussians as either static (3D) or dynamic (4D), enabling an adaptive strategy that optimizes storage and computation.
- **Significantly reduced training time.** By removing redundant temporal parameters for static Gaussians, our approach converges about 3–5× faster than baseline 4DGS methods while preserving fidelity.
- **Memory efficiency.** Converting large static regions to 3D Gaussians lowers memory requirements, allowing longer sequences or more detailed scenes given the same hardware specification.
- **High-fidelity dynamic modeling.** Focusing time-variant parameters on genuinely dynamic content achieves comparable or superior visual quality to 4DGS only representations across various challenging scenes.

## 2 RELATED WORK

### 2.1 NOVEL VIEW SYNTHESIS

The field of novel view synthesis has transitioned from fully implicit neural fields to more explicit representations that enable faster training and rendering. Neural Radiance Fields (NeRF) (Mildenhall et al., 2021) introduced the foundational approach by modeling scenes as continuous volumet-

ric functions from multi-view images. However, its reliance on deep MLP weights results in slow training and rendering times, motivating extensive research into more efficient alternatives. A key development in this direction involves replacing fully implicit representations with voxel grids, hash-encodings, or compact tensor-based structures (Müller et al., 2022; Sun et al., 2022b; Fridovich-Keil et al., 2022; Chen et al., 2022; Nam et al., 2023; Sun et al., 2022a; Fridovich-Keil et al., 2023; Barron et al., 2021). These approaches significantly reduce computational overhead by using spatially structured representations, enabling near-real-time rendering while maintaining high reconstruction fidelity.

More recently, point-based approaches have emerged as a promising alternative, culminating in *3D Gaussian Splatting* (3DGS) (Kerbl et al., 2023), which represents a scene as a collection of anisotropic Gaussian primitives.optimizing 3DGS for broader scalability presents challenges in memory efficiency and training speed. In terms of compact representations, several methods have explored utilizing vector quantization (Lee et al., 2024; Navaneet et al., 2023; Wang et al., 2024; Niedermayr et al., 2024; Papantonakis et al., 2024), entropy coding (Chen et al., 2024), and image or video codes (Morgenstern et al., 2024; Lee et al., 2025b). Regarding fast training, Mini-Splatting2 (Fang & Wang, 2024) and Turbo-GS (Lu et al., 2024b) demonstrate that near-minute training times are feasible via aggressive densification and careful tuning, suggesting that 3DGS can be optimized far more quickly with the right strategies.

## 2.2 Dynamic Scene Representation

Dynamic scene reconstruction extends static modeling techniques to time-varying objects and environments. Early works, such as D-NeRF (Pumarola et al., 2021) and Neural Volumes (Lombardi et al., 2019), used time-conditioned radiance fields to track temporal changes, enabling the representation of dynamic objects and their interactions over time. More recent methods based on explicit representations (Fridovich-Keil et al., 2023; Cao & Johnson, 2023; Fang et al., 2022; Shao et al., 2023; Song et al., 2023) decompose 4D scenes into lower-dimensional spaces, providing efficient ways to capture spatial and temporal dynamics both while improving scalability and rendering performance.

Building upon 3DGS, extended methods (Yang et al., 2023a; Duan et al., 2024; Li et al., 2024; Lee et al., 2025a) represent scenes with 4D Gaussian primitives, incorporating space-time geometry and corresponding features for real-time dynamic content rendering. Other approaches (Luiten et al., 2024; Yang et al., 2023b) model motion through 6-DoF trajectories or deformation fields, learning to transform Gaussians between frames. However, treating every scene component as dynamic can be inefficient, especially when the background remains static while only certain components move.

Our approach leverages the insight that modeling the entire scene with dynamic components is inefficient. We distinguish between static and dynamic content by introducing a novel scale-based classification method to automatically identify static regions, improving training and rendering speed, memory efficiency, and achieving performance on par with existing state-of-the-art methods for dynamic novel view synthesis.

## 3 Preliminary

In this section, we provide an overview of 3D Gaussian Splatting (3DGS) and its extension to dynamic scenes, 4D Gaussian Splatting (4DGS), which serve as the foundation for our approach.

### 3.1 3D Gaussian Splatting

3D Gaussian Splatting (3DGS) represents a scene by optimizing a collection of anisotropic 3D Gaussian ellipsoids, each defined by its center position $\mu$, and covariance matrix $\Sigma$, which encodes spatial extent and orientation:

$$G(x) = \exp\left(-\frac{1}{2}(x - \mu)^\top \Sigma^{-1}(x - \mu)\right), \tag{1}$$

where $x$ denotes a point in 3D space. To impose a structured representation, the covariance matrix $\Sigma$ is reparameterized using a rotation matrix $R$ and a scaling matrix $S$:

$$\Sigma = R\,S\,S^\top R^\top, \tag{2}$$

where, $S$ controls the scaling along the principal axes, and $R$ defines the orientation. Rendering is performed via alpha compositing, aggregating Gaussian contributions per pixel:

$$C = \sum_{i \in \mathcal{N}} c_i \alpha_i \prod_{j=1}^{i-1} (1 - \alpha_j), \tag{3}$$

where $c_i$ and $\alpha_i$ denote the color and opacity of the $i$-th Gaussian, and $\mathcal{N}$ denotes a set of Gaussians affecting a pixel to be rendered. This approach ensures a smooth and realistic blending of overlapping Gaussian contributions.

### 3.2 4D GAUSSIAN SPLATTING

Dynamic scene modeling requires extending the 3D formulation to model the temporal variations. 4D Gaussian Splatting (4DGS) (Yang et al., 2023a) achieves this by incorporating an additional temporal dimension into the 3D Gaussian representation, enabling the capture of motion and scene changes over time.

In the 4DGS framework, the spatial and temporal components are jointly modeled, resulting in four-dimensional rotation matrix, formulated as follows,

$$R = R_l\,R_r = \begin{bmatrix} a & -b & -c & -d \\ b & a & -d & c \\ c & d & a & -b \\ d & -c & b & a \end{bmatrix} \begin{bmatrix} p & -q & -r & -s \\ q & p & s & -r \\ r & -s & p & q \\ s & r & -q & p \end{bmatrix} \tag{4}$$

where $R_l$ and $R_r$ are left and right rotation matrix, each constructed by a quaternion vector, $(a, b, c, d)$ and $(p, q, r, s)$.

The temporally conditioned mean and covariance for a given time $t$ is computed as,

$$\mu_{xyz|t} = \mu_{1:3} + \Sigma_{1:3,4}\Sigma_{4,4}^{-1}(t - \mu_t), \tag{5}$$

$$\Sigma_{xyz|t} = \Sigma_{1:3,1:3} - \Sigma_{1:3,4}\Sigma_{4,4}^{-1}\Sigma_{4,1:3}. \tag{6}$$

For further details, please refer to the original 4DGS paper (Yang et al., 2023a).

## 4 HYBRID 3D-4D GAUSSIAN SPLATTING

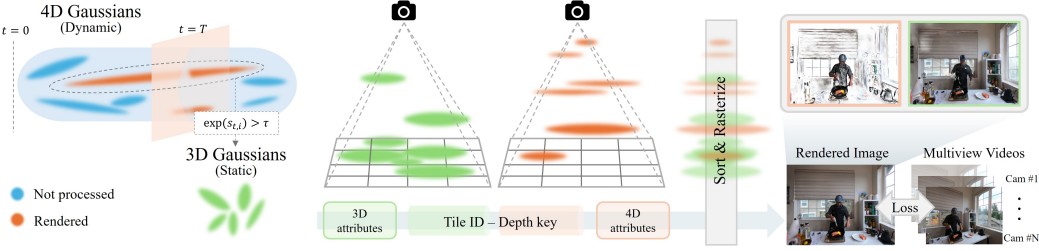

(a) 4D to 3D Conversion          (b) Hybrid Rendering and Training Pipeline

Figure 1: Overview of our hybrid 3D–4D Gaussian Splatting framework. (a) 4D Gaussians are optimized over time, and those exceeding a temporal scale threshold ($\tau$) are converted into 3D Gaussians. (b) Both 3D and 4D Gaussians are projected into screen space, assigned tile and depth keys, and sorted for rasterization. The rendered image is generated by blending static (3D) and dynamic (4D) Gaussians.

## 4.1 STATIC AND DYNAMIC REGION IDENTIFICATION

The prior works (Lee et al., 2025a; Liu et al., 2024) often identify static and dynamic content by analyzing the flow of Gaussians. Since our approach does not explicitly model the flows of 3D Gaussians, we leverage a 4D coordinate system, where each Gaussian has a scale parameter along the time axis. Concretely, each Gaussian is initially modeled as a 4D Gaussian, and for $i$-th Gaussian, its effective time-axis scale is given by $\exp(s_{t,i})$, where $\exp(\cdot)$ is an exponential activation function and $s_{t,i} \in \mathbb{R}$ denotes the time-axis scale parameter for $i$-th Gaussian. If $\exp(s_{t,i})$ exceeds a predefined threshold $\tau$, the Gaussian is classified as static Gaussian.

Intuitively, a larger temporal scale indicates that the Gaussian covers a static part of the scene without high-frequency temporal changes. Once a Gaussian's scale surpasses $\tau$, it is converted from a 4D (spatio-temporal) Gaussian to a 3D (spatial only) Gaussian. Importantly, this classification is performed dynamically at each densification stage rather than in a one-off preprocessing step. In other words, a Gaussian can remain 4D during early iterations and later transition to 3D once it expands to a larger temporal size. By continuously applying this process, our method adaptively separates static background elements from dynamic elements throughout the optimization process.

## 4.2 3D–4D GAUSSIAN CONVERSION

We convert each 4D Gaussian to a 3D Gaussian by discarding its temporal component and preserving its spatial components. More specifically, a 4D Gaussian is characterized by a mean

$$\mu_{4D} = (\mu_x, \mu_t), \tag{7}$$

where $\mu_x \in \mathbb{R}^3$ represents the spatial center and $\mu_t \in \mathbb{R}$ encodes the temporal coordinate. In addition, each Gaussian maintains a $4 \times 4$ rotation matrix $R_{4D}$, which determines how the Gaussian is oriented in the joint spatio-temporal domain. In principle, $R_{4D}$ can mix spatial and temporal axes, allowing the Gaussian to "tilt" across time.

For *static* Gaussians (those spanning the entire sequence without localized time variation), $R_{4D}$ effectively operates as a block-diagonal transform: the top-left $3 \times 3$ sub-block is a pure spatial rotation, and the time dimension remains separate. Formally,

$$R_{4D} = \begin{pmatrix} R_{3D} & \mathbf{0} \\ \mathbf{0}^\top & 1 \end{pmatrix} \quad \text{(ideal static case)}, \tag{8}$$

where $R_{3D} \in SO(3)$ is an orthonormal $3 \times 3$ rotation matrix and $\mathbf{0}$ is a three-dimensional zero vector. While this ideal case rarely happens in practice, we observe that by retaining only $R_{3D}$ information does not significantly affect the training process.

The corresponding unit quaternion for $R_{3D}$ matrix, $q_{3D} = (w, x, y, z)$, is derived as follows:

$$
\begin{aligned}
w &= \tfrac{1}{2}\sqrt{1 + \operatorname{tr}(R_{3D})}, \\
x &= \frac{R_{3D}(3,2) - R_{3D}(2,3)}{4\,w}, \\
y &= \frac{R_{3D}(1,3) - R_{3D}(3,1)}{4\,w}, \\
z &= \frac{R_{3D}(2,1) - R_{3D}(1,2)}{4\,w},
\end{aligned}
\tag{9}
$$

where $\operatorname{tr}(\cdot)$ is a trace operator, and $R_{3D}(\cdot, \cdot)$ denotes an element of the $R_{3D}$ matrix given an index.

Next, the temporal component of the mean, $\mu_t$, is discarded, and the spatial mean $\mu_x$ is retained as the 3D position of the Gaussian. Since the Gaussian is static, its position no longer changes over time; it remains fixed at $\mu_x$ in every time step. Also, its appearance attributes–including opacity $\sigma$ and spherical harmonic (SH) color coefficients–remain unchanged since static content does not require time-dependent updates. Consequently, each converted 3D Gaussian is fully specified by $(\mu_x, q_{3D}, s_x, s_y, s_z, \sigma, \text{SH})$, where $q_{3D}$ provides the orientation and $s_x, s_y, s_z$ specify the ellipsoid's principal scales. By converting all time-invariant Gaussians in this manner, we eliminate their dependence on temporal variable $t$ and reduce the dimensionality of the model. Meanwhile, dynamic

Table 1: Quantitative comparison on the N3V dataset (Li et al., 2022), with PSNR as the primary evaluation metric. The best and second-best results are highlighted in bold and underlined, respectively. For training time, (*): measured on our machine equiped with an RTX 4090 GPU, †: from Lee et al. (2025a), and other numbers are adopted from the original papers.

| Method | coffee_martini | cook_spinach | cut_roasted_beef | flame_salmon | flame_steak | sear_steak | Average | Training Time | FPS | Storage |
|---|---|---|---|---|---|---|---|---|---|---|
| HyperReel (Attal et al., 2023) | 28.37 | 32.3 | 32.92 | 28.26 | 32.2 | 32.57 | 31.1 | 9 h† | 2 | 360 MB |
| NeRFPlayer (Song et al., 2023) | 31.53 | 30.56 | 29.35 | **31.65** | 31.93 | 29.13 | 30.69 | 6 h | 0.05 | 5.1 GB |
| K-Planes (Fridovich-Keil et al., 2023) | 29.99 | 32.6 | 31.82 | 30.44 | 32.38 | 32.52 | 31.63 | 1.8 h | 0.3 | 311 MB |
| MixVoxel-L (Wang et al., 2023) | **29.63** | 32.25 | 32.4 | 29.81 | 31.83 | 32.1 | 31.34 | 1.3 h | 38 | 500 MB |
| 4DGS (Yang et al., 2023a) | 28.33 | 32.93 | **33.85** | 29.38 | **34.03** | 33.51 | 32.01 | (5.5 h)* | 114 | 2.1 GB |
| 4DGaussian (Wu et al., 2024) | 27.93 | 32.87 | 30.96 | 29.33 | 32.84 | 32.44 | 31.06 | (30 m)* | 137 | **34 MB** |
| STG (Li et al., 2024) | 28.61 | 33.18 | 33.52 | 29.48 | 33.64 | 33.89 | 32.05 | 1.3 h† | 140 | 200 MB |
| 4D-RotorGS (Duan et al., 2024) | 28.6 | 32.9 | 31.39 | 28.82 | 32.9 | 32.65 | 31.21 | 1 h | **277** | 144 MB |
| Ex4DGS (Lee et al., 2025a) | 28.79 | 33.23 | 33.73 | 29.29 | 33.91 | 33.69 | 32.11 | 36 m (1 h 8 m)* | 121 | 115 MB |
| **Ours** | 28.86 | **33.3** | 33.73 | 29.38 | 33.79 | **34.45** | **32.25** | **(11 m 53 s)*** | 208 | 273 MB |

Gaussians retain their full 4D parameterization (including time-based transformations). At runtime, each static Gaussian remains identical across frames, whereas each dynamic Gaussian is computed conditioned on the current timestamp.

## 4.3 OPTIMIZATION AND RENDERING PIPELINE

Table 2: Quantitative comparison on the 40-second sequence. The best and second-best results are highlighted in bold and underlined, respectively. All metric scores are taken from Xu et al. (2024b). ‡: Initializes point clouds using sparse COLMAP from each frame, **: split all 300 frames for training.

| Method | PSNR ↑ | SSIM ↑ | LPIPS ↓ | Training Time | VRAM | FPS | Storage |
|---|---|---|---|---|---|---|---|
| ENeRF (Lin et al., 2022) | 23.48 | 0.8944 | 0.2599 | 4.6 h | 23 GB | 5 | 0.83 GB |
| 4K4D** (Xu et al., 2024a) | 21.29 | 0.8266 | 0.3715 | 26.6 h | 84 GB | 290 | 2.46 GB |
| Dy3DGS (Luiten et al., 2024) | 25.91 | 0.8809 | 0.2555 | 37.1 h | **5 GB** | 610 | 19.5 GB |
| 4DGS** (Yang et al., 2023a) | 28.89 | **0.9521** | 0.1968 | 10.4 h | 84 GB | 90 | 2.68 GB |
| Xu‡ (Xu et al., 2024b) | **29.44** | 0.945 | 0.2144 | 2.1 h | 6.1 GB | 550 | **0.09 GB** |
| **Ours** | 29.2 | 0.9175 | **0.1173** | **52 m** | 12 GB | 111 | 0.96 GB |

We perform a short initial training phase (up to 500 iterations) with the full 4DGS model, allowing the 4D Gaussians to stabilize. We then apply the proposed static/dynamic identification scheme to split 4DGS into two groups: 3D and 4D Gaussians. Alongside this process, we apply adaptive densification and pruning separately to 3D and 4D Gaussians (also every 100 iteration), ensuring continuous refinement within their respective optimization pipelines.

This split mechanism and separate optimization substantially accelerate the training. In the original 4DGS training, only a small subset of 4D Gaussians is updated per training iteration, as many are culled when they do not contribute significantly to the rendering of training image timesteps. On the other hand, our approach updates static 3D Gaussians in every training iteration, leading to much faster convergence. As a result, our model typically converges in approximately 6K iterations for 10-second dynamic scenes, whereas standard 4DGS methods often require 20K to 30K iterations to achieve comparable visual quality.

Additionally, we eliminate opacity resets during training, a technique commonly used in 3D Gaussian splatting piplines to remove floaters in static scenes. While effective for static reconstructions, we found that periodic opacity reinitialization disrupts joint spatial-temporal optimization in dynamic scenes, particularly when training time is limited. Instead, we opt for a straightforward continuous optimization in which both static and dynamic Gaussians retain their opacities throughout the training procedure, achieving more stable convergence. Furthermore, since our hybrid model inherently reduces the number of Gaussians, it mitigates opacity saturation issues without requiring resets, unlike standard static scene reconstruction methods.

Finally, we integrate both 3D and 4D Gaussians into a unified CUDA rasterization pipeline. Our method builds upon the original 3DGS implementation (Kerbl et al., 2023), extending it to support 4D Gaussians at arbitrary timestamps alongside static ones. As illustrated in Fig. 1, each 4D Gaussian is sliced at time $t$ to generate a transient 3D Gaussian with mean $\mu_{xyz|t}$ and covariance $\Sigma_{xyz|t}$. We then aggregate all Gaussians (both 3D and 4D) into a single list, project them into screen space, assign tile and depth keys, and sort them for back-to-front alpha compositing. By rendering both types of Gaussians in a single pass, our approach maintains the efficiency of 3D splatting while preserving the flexibility of 4D temporal modeling.

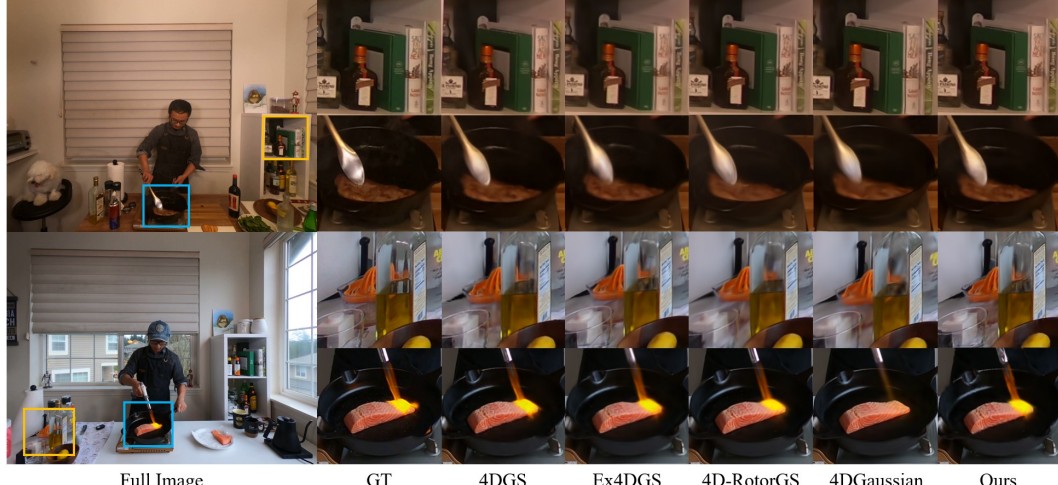

| Full Image | GT | 4DGS | Ex4DGS | 4D-RotorGS | 4DGaussian | Ours |

Figure 2: Qualitative comparison on the N3V dataset. While most methods yield comparable results, our approach can preserve subtle motion cues and slightly more consistent colors in some challenging regions. Zoom in for best viewing.

## 5 EXPERIMENTS

### 5.1 DATASETS

**Neural 3D Video (N3V).** We evaluate our method on the N3V dataset (Li et al., 2022), which comprises six multi-view video sequences captured using 18-21 cameras at a native resolution of $2704 \times 2028$. Five sequences last 10 seconds each, while one sequence spans 40 seconds. In most experiments, we follow standard practice by using 10-second segments for fair comparisons, specifically extracting a 10-second clip from the 40-second video (flame_salmon). In line with prior work, we hold out cam00 as the test camera for each scene and use the remaining cameras for training. Additionally, we experiment with the full 40-second sequence to demonstrate the scalability and robustness of our method on longer dynamic content. For all experiments, we downsample the videos by a factor of two (both training and evaluation), following the protocol used in previous works.

**Technicolor.** We also evaluate our method on a subset of the Technicolor dataset (Sabater et al., 2017), which comprises video recordings from a $4 \times 4$ camera array (16 cameras) at a resolution of 2048×1088. Following the common practice, we select five scenes (Birthday, Fabien, Painter, Theater, Trains), each limited to 50 frames. We keep the original resolution and designate cam10 as the held-out test view, using the remaining cameras for training.

### 5.2 IMPLEMENTATION DETAILS

Table 3: Quantitative results on the Technicolor dataset. Training times (including COLMAP) are measured on the Painter scene with an RTX 3090 GPU. For training time, (∗): measured on our machine, †: from Bae et al. (2024), ‡: uses sparse COLMAP initialization.

| Method | PSNR ↑ | SSIM ↑ | LPIPS ↓ | Training Time | Storage |
|---|---|---|---|---|---|
| HyperReel (Attal et al., 2023) | 33.32 | 0.899 | 0.118 | 2 h 45 m† | 289 MB |
| 4DGaussian (Wu et al., 2024) | 29.62 | 0.844 | 0.176 | 32 m† | **51 MB** |
| E-D3DGS (Bae et al., 2024) | 33.24 | 0.907 | 0.100 | 3 h 02 m† | 77 MB |
| 4DGS (Yang et al., 2023a) | 33.35 | 0.910 | 0.095 | (4 h 20 m)∗ | 1.07 GB |
| Ex4DGS‡ (Lee et al., 2025a) | **33.62** | **0.9156** | **0.088** | (1 h 5 m)∗ | 88 MB |
| **Ours**‡ | 33.22 | 0.911 | 0.149 | **(29 m)**∗ | 218 MB |

Following Yang et al. (2023a), we initialize our 4D Gaussian representation using dense COLMAP reconstructions for the N3V dataset (about 300k points), providing robust geometric priors. For Technicolor, which has only 50 frames per scene, we start from a sparse COLMAP reconstruction instead. We adopt the densification pipeline from 3D Gaussian Splatting Kerbl et al. (2023), progressively increasing the number of Gaussians by cloning and splitting operations. Unlike prior works, however, we do not perform periodic opacity resets during training. For automatic classification of Gaussians, we set the temporal scale threshold $\tau$ to 3 for the 10-second N3V sequences and 6 for the 40-second sequence, while using a threshold of 1 for Technicolor. We train the 10-second N3V clips for 6,000 iterations (batch size 4) and the 40-second clip for 20,000 iterations, applying the

adaptive densification up to 15,000 iterations. For Technicolor, each scene is trained for 10,000 iterations with a batch size of 2. Our implementation is built on the codebase of Yang et al. (2023a) and further leverages the efficient backward pass from Taming-3DGS Mallick et al. (2024) to accelerate optimization.

## 5.3 RESULTS

### 5.3.1 QUANTITATIVE RESULTS

**N3V Dataset.** We first evaluate our approach on the N3V dataset, with results summarized in Tab. 1. Our method achieves competitive performance across all scenes, with an average PSNR of 32.25 dB, outperforming recent methods in both fidelity and rendering speed. Notably, we require only 12 minutes of training time for the 10-second clips, which is significantly faster than 4DGS (Yang et al., 2023a) (5.5 hours), while providing comparable or superior visual quality. The combination of fast optimization, high FPS (208), and moderate storage (273 MB) underscores the effectiveness of our hybrid 3D–4D Gaussian representation.

Table 4: Ablation study on the N3V dataset, comparing the 4DGS baseline, our approach (Ours), the effect of opacity resets (w/ opa reset), and different temporal scale thresholds ($\tau$). #4D and #3D denote the number of 4D and 3D Gaussians, respectively.

| Method | PSNR | SSIM | LPIPS | #4D | #3D |
|---|---|---|---|---|---|
| 4DGS (Yang et al., 2023a) | 32.01 | 0.9453 | 0.0974 | 3,315,333 | – |
| Ours | 32.25 | 0.9459 | 0.0970 | 843,175 | 229,707 |
| w/ opa reset | 31.52 | 0.9418 | 0.1016 | 683,437 | 243,051 |
| $\tau = 2.5$ | 31.37 | 0.9440 | 0.0979 | 670,807 | 276,265 |
| $\tau = 3.5$ | 31.98 | 0.9450 | 0.0986 | 913,927 | 184,548 |

**Long Sequence (40 seconds).** Tab. 2 presents the results on the challenging 40-second clip from the N3V dataset. Our method achieves the second-best PSNR (29.2 dB) and the lowest LPIPS (0.1173), demonstrating strong perceptual quality. Remarkably, we complete training in only 52 minutes, an order of magnitude faster than other methods. Although Xu et al. (2024b) reports a slightly higher PSNR (29.44 dB) by initializing point clouds from every frame (sparse COLMAP for each frame takes approximately 1 second, additional 20 minutes for 1,200 frames to their reported training time 2.1 hours), our approach relies solely on the single-frame initialization used for 10-second experiments. Despite this simpler setup, our method provides a more balanced trade-off in terms of training speed, storage, and inference performance, highlighting its scalability to longer sequences.

**Technicolor Dataset.** We further validate our method on the Technicolor dataset (Tab. 3). Despite using a sparse COLMAP initialization for the 50-frame sequences, our model achieves 33.22 dB PSNR and 0.911 SSIM, with only 29 minutes of training time on an RTX 3090 (measured on the `Painter` scene). In contrast, 4DGS requires over four hours to reach a comparable PSNR, and Ex4DGS—while slightly more accurate—needs more than twice of our training time. Our final storage is 218 MB, which is lower than 4DGS (1.07 GB) but slightly higher than some other methods. Overall, these results confirm that our framework effectively handles diverse camera setups and short videos, balancing speed, memory efficiency, and rendering fidelity.

(a) static          (b) dynamic

Figure 3: Visual comparison of different scale thresholds $\tau$.

### 5.3.2 QUALITATIVE RESULTS

Fig. 2 compares our method with several baselines on the N3V dataset. Overall, the visual quality among these methods is largely similar, reflecting the challenging nature of dynamic scenes. However, our hybrid representation shows sharper details in some dynamic regions and more consistent color transitions in backgrounds, reducing minor flickers across frames. These observations align with our quantitative findings, suggesting that our approach remains competitive for complex, real-world scenarios.

## 5.4 ABLATION STUDIES AND ANALYSIS

**Scale Threshold** $\tau$. We investigate how varying the temporal scale threshold $\tau$ affects both reconstruction quality and storage (see Tab. 4). As shown in Fig. 3, a lower threshold (e.g., $\tau = 2.5$) aggressively converts 4D Gaussians into 3D, which can inadvertently merge dynamic content into the static representation, reducing motion detail despite simplifying the final geometry. Conversely, a higher threshold ($\tau = 3.5$) is more lenient about switching Gaussians to 3D, preserving subtle dynamics at the cost of slower convergence and higher memory usage. The mid-range setting ($\tau = 3.0$) strikes a balanced trade-off, maintaining near-optimal quality while avoiding excessive storage overhead.

**Opacity Reset.** Many 3D/4D Gaussian methods periodically reinitialize opacities to a small constant to remove floaters or spurious elements (Kerbl et al., 2023; Yang et al., 2023a). However, such resets are heuristic and can inadvertently disrupt optimization in dynamic regions. As shown in Tab. 4 and Fig. 4, forcibly lowering the opacities of both 3D and 4D Gaussians can erase previously learned motion cues, leading to flicker or lower final PSNR. By avoiding opacity resets, our pipeline continuously refines all Gaussians in a single pass, preserving subtle temporal details and stabilizing motion boundaries. This simpler, reset-free approach also reduces hyperparameter tuning overhead and prevents abrupt representation changes that might otherwise degrade performance.

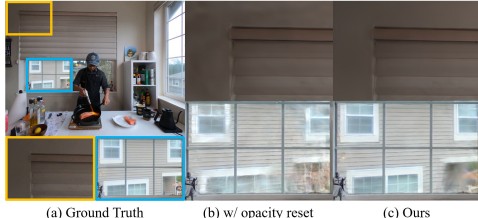

(a) Ground Truth  (b) w/ opacity reset  (c) Ours

Figure 4: Influence of opacity resets on a dynamic scene.

**Visualization of spatially distributed Gaussians** Fig. 5 visualizes the spatially distributed Gaussians, comparing our model to 4DGS (Yang et al., 2023a). To visualize, we first project all 3D and 4D Gaussians (for 4DGS, only 4D Gaussians) on the image plane given a specific view point. Then, we colorcoded based on the number of projected Gaussians in each spatial location (the darker color, the more Gaussians). This shows how each approach allocates Gaussians differently to different spatial regions, and the original 4DGS introduces many Gaussians in static areas (highlighted as red boxes), implying that numerous 4D Gaussians with small time scales are used to represent static parts of the scene. On the other hand, our approach uses 3D Gaussians for static areas, resulting in evenly distributed Gaussians across the scene. This result supports our experimental results that our method significantly reduces redundancy, lowers memory usage, and accelerates optimization. By contrast, the baseline model places dense clusters of Gaussians in static regions, leading to unnecessary computations, inflating memory costs, and often degrading the rendering quality.

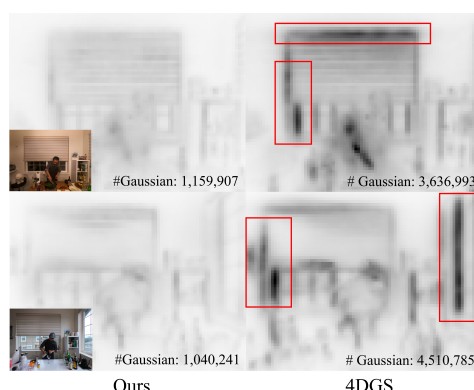

#Gaussian: 1,159,907  # Gaussian: 3,636,993

#Gaussian: 1,040,241  # Gaussian: 4,510,785

Ours  4DGS

Figure 5: Visualization of spatially distributed Gaussians.

## 6 CONCLUSION

We have presented a novel hybrid 3D-4D Gaussian Splatting framework for dynamic scene reconstruction. By distinguishing static regions and selectively assigning 4D parameters only to dynamic elements, our method substantially reduces redundancy while preserving high-fidelity motion cues. Extensive experiments on the N3V and Technicolor datasets demonstrate that our approach consistently achieves competitive or superior quality and faster training compared to state-of-the-art baselines.

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

# Supplementary materials

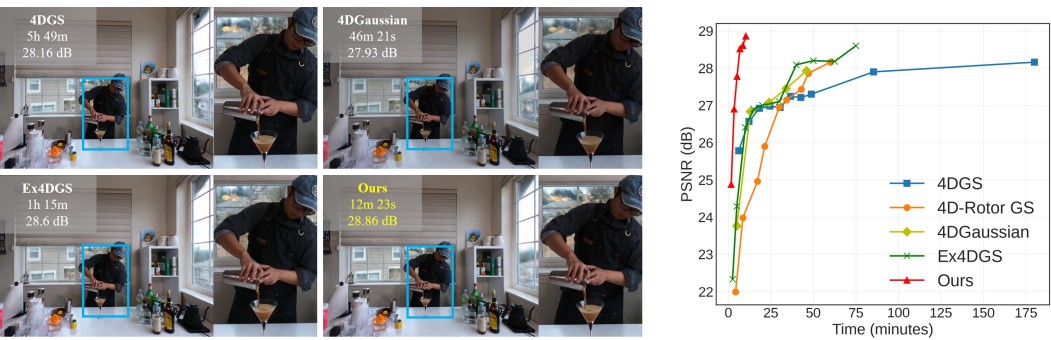

Figure 6: **Left:** Rendering results on the coffee_martini scene. **Right:** PSNR vs. training time. The proposed method converges in 12 minutes while maintaining competitive rendering quality. All methods were evaluated under the same machine equipped with the NVIDIA RTX4090 GPU, except for 4D-Rotor GS Duan et al. (2024)—whose results were estimated from iteration counts since the code is not publicly available.

## A  ANALYSIS OF TEMPORAL SCALE ON 4DGS

As shown in Fig. 7, the majority of Gaussians in a fully trained 4DGS (Yang et al., 2023a) model have small temporal scales (typically below 0.5), which results in redundant memory consumption and increased computational cost. We empirically set the threshold $\tau$ by analyzing the distribution of temporal scales in 4DGS and considering the characteristics of our target datasets. Specifically, $\tau$ is selected to fall within the "valley" that separates lower (more dynamic) and higher (more static) scale values.

## B  CUDA RASTERIZATION PIPELINE

Algorithm 1 represents our rasterization process. Compared to the original pipeline in the 3DGS (Kerbl et al., 2023), lines 4–6 are newly introduced to seamlessly integrate static (3D) Gaussians with dynamic (4D) Gaussians. In particular, the size of $M'$ is allocated to accommodate both 3D and 4D points. The conditional check at line 4 verifies whether any 3D Gaussians exist; if so, it projects them into screen space via ProjGaussian3D, and stores tile, depth, and screen-space position data jointly with the 4D Gaussians.

## C  ADDITIONAL RESULTS

As shown in Fig. 6, we achieved near state-of-the-art reconstruction fidelity while substantially reducing training time compared to prior 4DGS baselines. We also provide further quantitative and qualitative evaluations to supplement our main paper in this section.

### C.1  SSIM AND LPIPS COMPARISONS

We present additional metrics on SSIM and LPIPS for the N3V dataset. As summarized in Table 5, our method consistently maintains strong perceptual quality across these metrics, corroborating the PSNR improvements reported in the main text. In particular, our SSIM and LPIPS scores remain on par with, or exceed, those of baseline methods, indicating sharper details and fewer artifacts in dynamic regions.

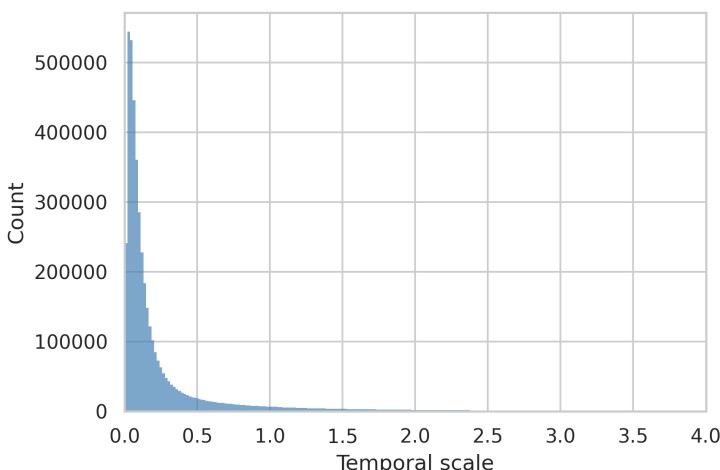

Figure 7: Distribution of the t-axis scale for Gaussians in the `coffee_martini` scene. Most Gaussians cluster at smaller scales, indicating dynamic content, while a minority have larger scales that suggest static regions.

---

**Algorithm 1** GPU Rasterization of 3D&4D Gaussians

---

**Require:** $w, h$: image dimensions
**Require:** $M_{4D}, S_{4D}$: 4D Gaussian means and covariances
**Require:** $M_{3D}, S_{3D}$: 3D Gaussian means and covariances
**Require:** $A$: 3D/4D Gaussian attributes
**Require:** $V$: camera/view configuration
**Require:** $s$: time
 1: **function** RASTERIZE($w, h, M_{4D}, S_{4D}, M_{3D}, S_{3D}, A, V, s$)
 2:     CullGaussian($p, V$)
 3:     $(M', S'_{4D}) \leftarrow$ ProjGaussian4D($M_{4d}, S_{4d}, V, s$)
 4:     **if** $len(M_{3D}) > 0$ **then**
 5:         $(M', S'_{3d}) \leftarrow$ ProjGaussian3D($M', M_{3d}, S_{3d}, V$)
 6:     **end if**
 7:     $T \leftarrow$ CreateTiles($w, h$)
 8:     $(L, K) \leftarrow$ DuplicateWithKeys($M', T$)
 9:     SortByKeys($K, L$)
10:     $R \leftarrow$ IdentifyTileRanges($T, K$)
11:     $I \leftarrow 0$
12:     **for all** Tiles $t \in I$ **do**
13:         **for all** pixels $i \in t$ **do**
14:             $r \leftarrow$ GetTileRange(R,t)
15:             $I[i] \leftarrow$ BlendInOrder($i, L, r, K, M', S'_{4D}, S'_{3D}, A$)
16:         **end for**
17:     **end for**
18:     **return** $I$
19: **end function**

---

### C.2 PER-SCENE GRAPHS ON N3V

Fig. 8 shows the per-scene PSNR curves over training iterations for three different scale thresholds. While $\tau = 2.5$ can converge quickly in the early iterations, it sometimes saturates at a slightly lower peak PSNR (e.g., `cook_spinach`) or collapse after few iteration(e.g. `flame_steak`), possibly merging subtle dynamics into static representation. In contrast, $\tau = 3.5$ tends to retain more 4D Gaussians longer, occasionally surpassing $\tau = 2.5$ in later stages (e.g., `sear_steak`), but it also requires more training to reach its final quality. The mid-range threshold ($\tau = 3.0$) typically offers a balanced trade-off between these extremes, achieving stable and competitive performance across scenes with moderate or complex motion.

Table 5: Additional SSIM and LPIPS results on the N3V dataset. Higher SSIM and lower LPIPS indicate better perceptual quality.

| Method | SSIM ↑ | LPIPS ↓ |
|---|---|---|
| HyperReel Attal et al. (2023) | 0.927 | 0.096 |
| NeRFPlayer Song et al. (2023) | 0.931 | 0.111 |
| K-Planes Fridovich-Keil et al. (2023) | 0.947 | 0.090 |
| MixVoxel-L Wang et al. (2023) | 0.933 | 0.095 |
| 4DGS Yang et al. (2023a) | 0.9453 | 0.0974 |
| STG Li et al. (2024) | 0.948 | 0.046 |
| 4DGaussian Wu et al. (2024) | 0.935 | 0.074 |
| 4D-RotorGS Duan et al. (2024) | 0.939 | 0.106 |
| Ex4DGS Lee et al. (2025a) | 0.940 | 0.048 |
| **Ours** | 0.9459 | 0.097 |

## C.3 ADDITIONAL QUALITATIVE RESULTS

Finally, we present further visual comparisons, highlighting subtle differences in dynamic objects, complex lighting, and motion boundaries. Our hybrid 3D–4D representation consistently captures both static and moving elements with minimal artifacts, reinforcing the quantitative gains reported in the main paper.

**Long-Sequence Comparison.** In Fig. 9, we compare our reconstructions to ground-truth frames from the 40-second N3V sequence. Despite the longer duration and more complex motion, our method maintains coherent geometry and color transitions, demonstrating robust performance for extended temporal dynamics without significant artifacts.

**Multi-Dataset Visuals.** Fig. 10 showcases additional results on both N3V and Technicolor scenes. We observe that our method preserves fine-grained details under challenging lighting conditions, while effectively modeling diverse motion patterns. These qualitative improvements align with our quantitative gains in PSNR and SSIM.

**Dynamic and Static Visuals.** In Fig. 11, we visualize dynamic and static Gaussians side by side, with dynamic regions rendered on a white background to highlight the separation from static areas. Our method adaptively assigns 4D Gaussians to genuinely moving objects while converting large, motionless regions to 3D Gaussians. This selective allocation preserves subtle motion cues, reduces memory overhead, and accelerates the optimization process. The final rendered results confirm that our representation remains faithful to the original scenes, even under challenging lighting and motion conditions.

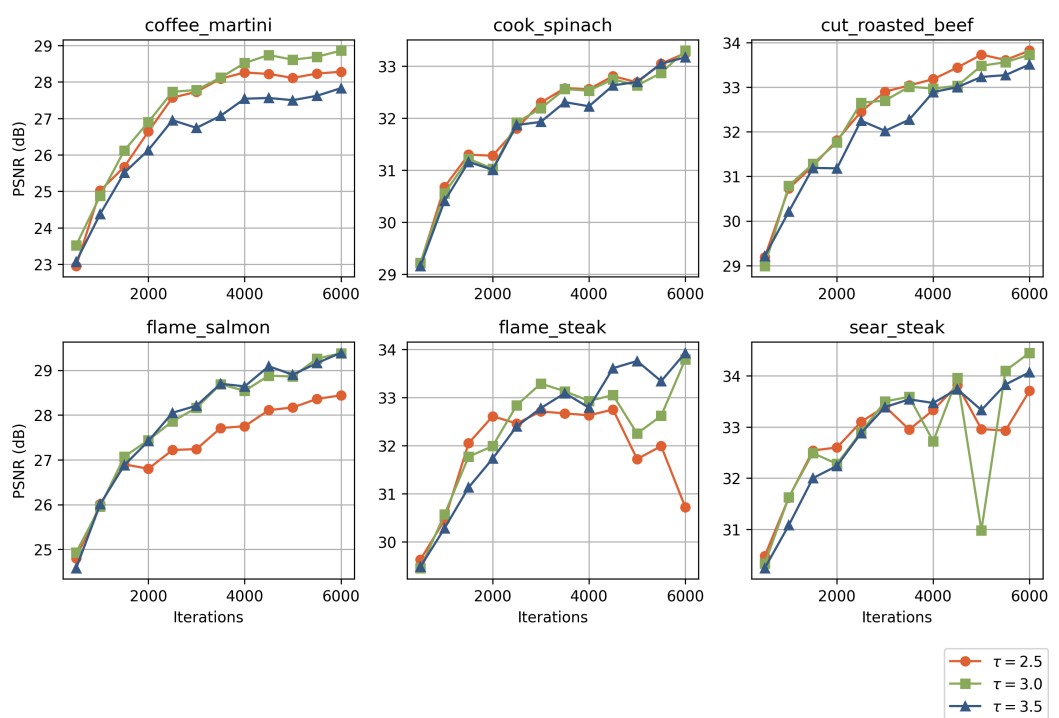

Figure 8: **Per-scene PSNR curves on the N3V dataset** for different temporal scale thresholds ($\tau = 2.5, 3.0, 3.5$). Each plot corresponds to one scene, showing how PSNR evolves over 6000 iterations of training. The mid-range setting ($\tau = 3.0$) often strikes a balance, maintaining competitive final quality across a range of motion complexities.

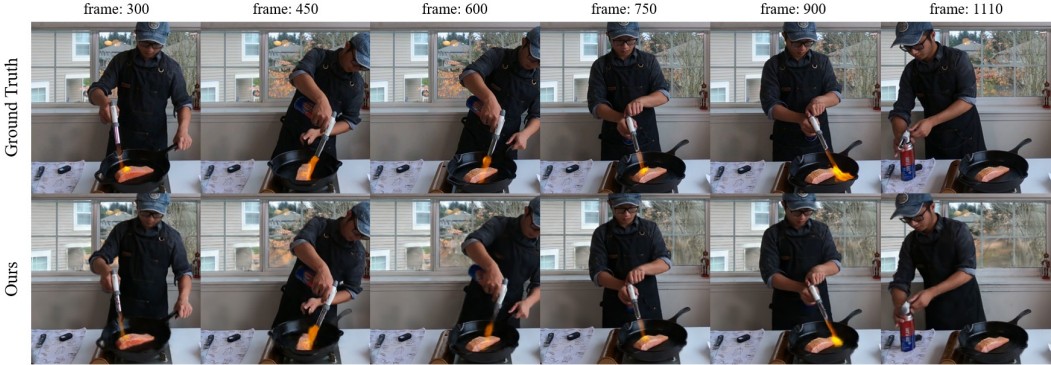

Figure 9: **Comparison with Ground Truth on the 40-second sequence.** We sample frames at different timestamps (top: GT, bottom: ours) to illustrate that our approach preserves both global structure and subtle motion details over extended temporal ranges.

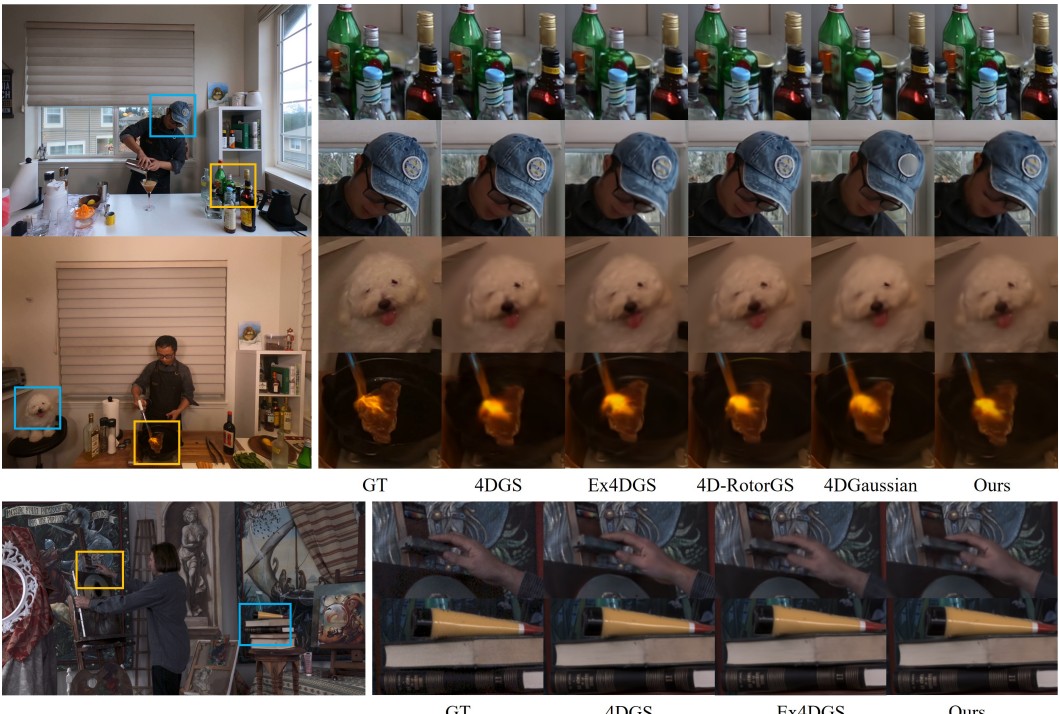

GT     4DGS     Ex4DGS     4D-RotorGS     4DGaussian     Ours

GT     4DGS     Ex4DGS     Ours

Figure 10: **Additional results on N3V and Technicolor scenes.** Despite challenging lighting conditions and fast motion, our hybrid 3D-4D approach maintains crisp object boundaries and more consistent textures across frames.

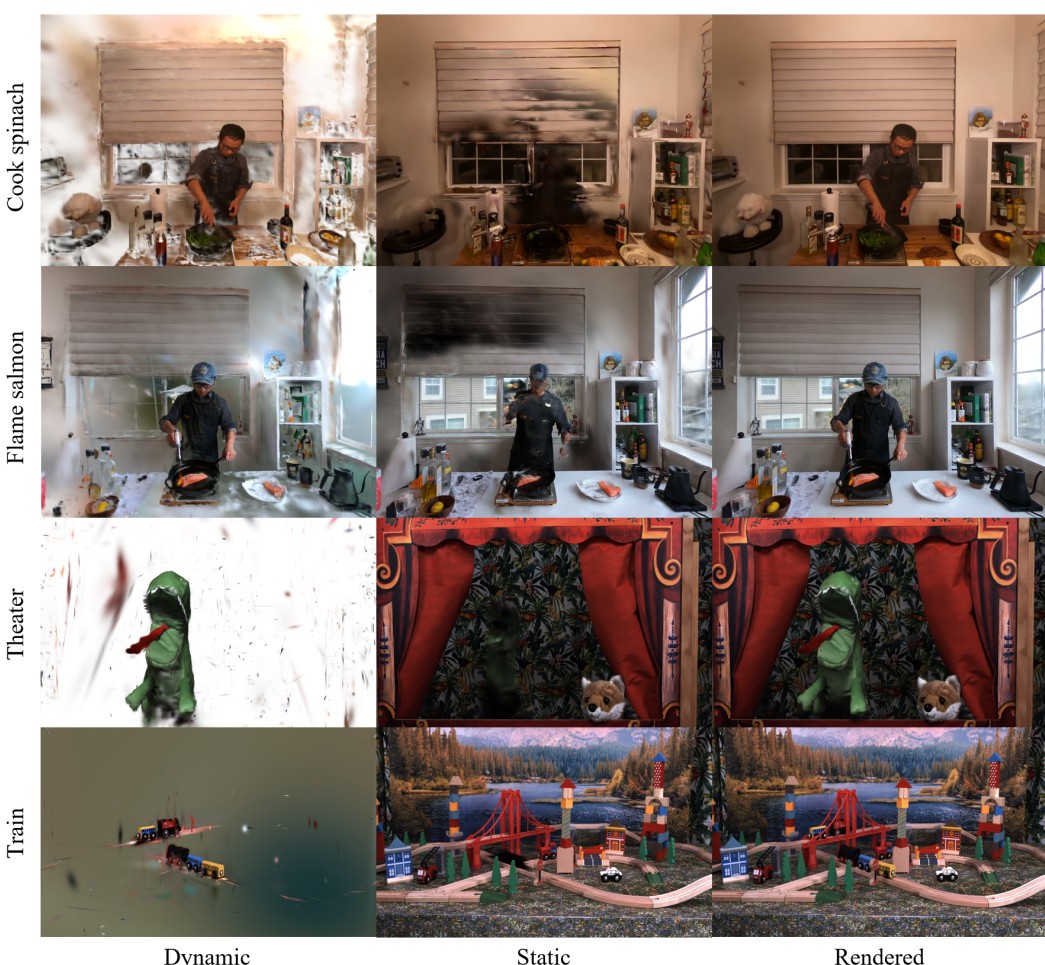

Figure 11: **Dynamic vs. Static Visualization.** Each row shows (left) the dynamic portion on a white background, (middle) the static region, and (right) the fully rendered result. By converting most static elements into 3D Gaussians, our approach effectively handles dynamic scenes while reducing redundant computations and preserving high-fidelity details.

