# OpenReview forum: "Hybrid 3D-4D Gaussian Splatting for Fast Dynamic Scene Representation"
_ICLR.cc/2026/Conference — ICLR 2026 Conference Withdrawn Submission_

### Official Review · Reviewer_v9sY · 2025-10-16

**Soundness:** 3
**Presentation:** 3
**Contribution:** 2
**Rating:** 2
**Confidence:** 5

**Summary:**

This paper introduces a novel Hybrid 3D-4D Gaussian Splatting (3D-4DGS) framework for dynamic scene representation. It adaptively identifies static and dynamic regions by initially representing all Gaussians in 4D and iteratively converting temporally invariant Gaussians into 3D ones. This approach significantly reduces computational and memory overhead

**Strengths:**

- Adaptive Hybrid Representation: Dynamically classifies Gaussians to balance computational cost and fidelity by modeling static regions in 3D and dynamic regions in full 4D.
- Fast Training and Memory Efficiency: Removes redundant temporal parameters for static Gaussians, achieving significantly shorter training times and reduced memory consumption without sacrificing quality.
- High-Fidelity Dynamic Modeling: Retains full 4D Gaussians for genuinely dynamic content, enabling accurate capture of complex motions with high perceptual quality.

**Weaknesses:**

- The proposed method tends to train many dynamic 4D Gaussians with very small temporal variation scales, which effectively leads to overparameterization. This structural characteristic allows the model to **"memorize" visual content of the scene rather than truly learning meaningful motion dynamics.**
The observed acceleration in training speed can ironically be interpreted as evidence of this overparameterization; instead of efficient spatiotemporal modeling, the network is fitting redundant parameters that do not contribute to genuine motion understanding.

- Moreover, Gaussian Splatting’s original strength lies in its explicit representation that naturally supports motion tracking and deformation modeling. However, this approach abandons that advantage by not explicitly modeling motion or deformation, **thus sacrificing the core benefit of explicit tracking**.

- Moreover, since this approach **essentially memorizes** only what is visually observed, I believe it will perform even worse in monocular settings where motion cues are sparse and ambiguous. Therefore, I remain quite skeptical about its effectiveness in such scenarios.

**Questions:**

N/A

---

### Official Review · Reviewer_uc5o · 2025-10-29

**Soundness:** 2
**Presentation:** 2
**Contribution:** 2
**Rating:** 4
**Confidence:** 4

**Summary:**

This paper aims to address the high computational and memory overhead associated with 4D Gaussian Splatting (4DGS) for dynamic 3D scene representation. The authors propose a novel hybrid 3D-4D Gaussian Splatting (3D-4DGS) framework. The core idea is to adaptively represent static regions using 3D Gaussians while reserving 4D Gaussians for dynamic elements. Experimental results indicate that the proposed method achieves significant training speedups and reduces memory consumption, all while maintaining or improving visual quality.

**Strengths:**

1. The paper addresses a practical and significant problem in the 4DGS domain: the redundant representation of static regions, which leads to high computational and memory overhead.

2. Compared to the 4DGS baseline, the proposed method shows improvements in training speed, storage, and reconstruction quality.

**Weaknesses:**

1. The core mechanism for distinguishing static from dynamic content relies on the temporal scale $s_t$ and a hard threshold $\tau$, which is fundamentally a heuristic approach. This threshold $\tau$ requires manual tuning for different datasets and sequence lengths (e.g., 3.0 for N3V 10s, 6.0 for 40s, and 1.0 for Technicolor). This suggests the hyperparameter may be highly sensitive to the scene's motion characteristics and lacks generalizability.

2. The improvements appear marginal. As shown in Table 1, the average PSNR reconstruction metric only improved from 32.01 to 32.25 compared to the baseline, which is not a significant gain. Furthermore, Table 3 indicates that the proposed method is outperformed by Ex4DGS in terms of both reconstruction quality and storage.

**Questions:**

1. On the adaptivity of $\tau$: The threshold $\tau$ appears to be a critical and sensitive hyperparameter that requires adjustment based on the dataset and even the sequence length. How did you determine the optimal $\tau$ value for each experiment? Have you considered any adaptive or automatic strategies for setting $\tau$ to enhance generalizability?

2. On the uni-directional conversion: The conversion from 4D to 3D appears to be uni-directional. The paper states, "Since the Gaussian is static, its position no longer changes over time; it remains fixed at $\mu_x$ in every time step." How would the model handle a scenario where a region is static during the initial training phase (and thus converted to 3D) but begins to move later in the sequence (e.g., a previously stationary object is picked up)?

---

### Official Review · Reviewer_MaUD · 2025-10-31

**Soundness:** 3
**Presentation:** 3
**Contribution:** 3
**Rating:** 4
**Confidence:** 5

**Summary:**

The paper introduces a refinement aimed at reducing memory usage and training time in 4D Gaussian Splatting by identifying static regions in a scene and representing them with 3D Gaussians. Traditional 4D Gaussian Splatting employs Gaussians in 4D space, slicing them into 3D Gaussians based on conditional time. However, since most areas in a scene remain static, this approach leads to inefficiency, as static regions are redundantly represented by multiple segments of 4D Gaussians. The proposed method addresses this redundancy by analyzing the temporal scale to detect static regions and converting the corresponding 4D Gaussians into fixed 3D Gaussians. As a result, these Gaussians remain constant, eliminating the need for additional 4D Gaussians to represent static parts. Experimental results demonstrate that the proposed method significantly reduces memory consumption and accelerates the training process.

**Strengths:**

The paper provides a clear and important insight: modeling a hybrid scene containing both static and dynamic components should utilize a hybrid representation, such as 3D-4D Gaussian Splatting. This is an underexplored problem that the authors have identified and effectively addressed. Previous approaches, including 4DGS, StreamGS, and STGS, rely on temporal Gaussians to represent the entire scene, overlooking the distinction between static and dynamic regions. This paper highlights this gap and offers a plausible solution.

Experimental results demonstrate that the proposed method not only significantly reduces training time but also achieves superior rendering quality compared to 4DGS, on which it is based.

**Weaknesses:**

1. The hyperparameter \(\tau\), which controls the identification of static Gaussians, was carefully tuned. As shown in Table 4, even a slightly higher or lower value results in performance degradation, making the method perform worse than 4D Gaussians. This highlights a lack of robustness in the approach. For scenes with varying characteristics, such as differing ratios of dynamic components or varying motion amplitudes, \(\tau\) would need to be adjusted accordingly. If not properly selected, the performance drops significantly, indicating a critical dependency on this parameter.

2. The advantages of the method in rendering quality are not particularly evident. Both the quantitative evaluations and visual comparisons of rendered images show only marginal improvements, if any. In fact, Ex4DGS outperforms the proposed method in both file size and rendering quality. However, since the primary goal of the method is to improve 4DGS by enabling faster training and reducing storage requirements, this limitation is somewhat acceptable. That said, Ex4DGS is only about three times slower in training while achieving better performance and a smaller file size, which still makes this a notable weakness of the proposed approach.

**Questions:**

1. Could you clarify how robust the default value of \(\tau = 3\) is? How does the method perform when applied to scenes with varying dynamic characteristics, such as differing motion amplitudes or dynamic ratios? Additional experiments focusing on such scenarios could provide more convincing evidence of the method's robustness.

2. What specific advantages does the proposed method offer over Ex4DGS? The only notable benefits seem to be faster rendering speed and reduced training time. However, these advantages are relatively limited according to the quantitative evaluation. The most critical metrics, rendering quality and storage efficiency, are the core focus of the paper, yet the method underperforms compared to Ex4DGS in these areas. This raises concerns about the overall effectiveness of the proposed approach.

---

### Official Review · Reviewer_5c84 · 2025-10-31

**Soundness:** 2
**Presentation:** 2
**Contribution:** 2
**Rating:** 4
**Confidence:** 4

**Summary:**

This paper proposes a hybrid 3D-4DGS representation for dynamic scene reconstruction. The core idea of this paper is that by converting 4DGS with large time-axis scales to 3DGS, the hybrid 3D-4DGS presentation can achieve better performance on dynamic scene reconstruction.

**Strengths:**

1.This paper is well-written and easy to follow.
2.The idea of modeling the dynamic and static parts of a scene separately is very intuitive.

**Weaknesses:**

1.Lack of Novelty. The representations for modeling dynamic and static objects, 3D/4DGS, are borrowed from existing works, and the conversion between them is simply the result of discarding the temporal dimension. There is no unique insight.
2.Subtle Performance Improvements. The quantitative improvements compared to existing SOTA methods (even the baseline 4DGS) are limited. The differences are also difficult to be seen in the qualitative comparison.
3.Evaluations on Decoupling Static/Dynamic Elements. Since the core idea of this paper is the separate modeling of dynamic and static scenes, it would be more convincing to conduct both quantitative and qualitative comparisons with baselines in tasks such as distractor-free scene reconstruction or static scene reconstruction from dynamic videos (e.g., DAS3R[1], WildGaussians[2]).

[1] Xu, Kai, et al. "Das3r: Dynamics-aware gaussian splatting for static scene reconstruction." arXiv preprint arXiv:2412.19584 (2024).
[2] Kulhanek, Jonas, et al. "WildGaussians: 3D Gaussian Splatting In the Wild." Advances in Neural Information Processing Systems 37 (2024): 21271-21288.

**Questions:**

1.In this paper, the conversion from dynamic 4DGS to static 3DGS appears to be an irreversible process. This seems to easily lead to static 3DGS incorrectly representing the dynamic parts of the scene, thus affecting the reconstruction and rendering results.

---

### Comment · Area_Chair_CAzr · 2025-11-25
**Please check and engage with the reviewers (if the authors plan to)**

Dear authors,

It has been over one week since the review comments were released.

If you plan to join the rebuttal, please engage sooner rather than later, so that the reviewers can have time engage as well.

Thanks,
AC

---

### Note · Authors · 2025-11-27

I have read and agree with the venue's withdrawal policy on behalf of myself and my co-authors.